# Is Standardization Necessary for Sharing of a Large Mid-Infrared Soil Spectral Library?

**DOI:** 10.3390/s20236729

**Published:** 2020-11-25

**Authors:** Shree R. S. Dangal, Jonathan Sanderman

**Affiliations:** Woodwell Climate Research Center, Falmouth, MA 02540, USA

**Keywords:** mid-infrared diffuse reflectance spectroscopy, calibration transfer, piecewise direct standardization (PDS), soil properties

## Abstract

Recent developments in diffuse reflectance soil spectroscopy have increasingly focused on building and using large soil spectral libraries with the purpose of supporting many activities relevant to monitoring, mapping and managing soil resources. A potential limitation of using a mid-infrared (MIR) spectral library developed by another laboratory is the need to account for inherent differences in the signal strength at each wavelength associated with different instrumental and environmental conditions. Here we apply predictive models built using the USDA National Soil Survey Center–Kellogg Soil Survey Laboratory (NSSC-KSSL) MIR spectral library (*n* = 56,155) to samples sets of European and US origin scanned on a secondary spectrometer to assess the need for calibration transfer using a piecewise direct standardization (PDS) approach in transforming spectra before predicting carbon cycle relevant soil properties (bulk density, CaCO_3_, organic carbon, clay and pH). The European soil samples were from the land use/cover area frame statistical survey (LUCAS) database available through the European Soil Data Center (ESDAC), while the US soil samples were from the National Ecological Observatory Network (NEON). Additionally, the performance of the predictive models on PDS transfer spectra was tested against the direct calibration models built using samples scanned on the secondary spectrometer. On independent test sets of European and US origin, PDS improved predictions for most but not all soil properties with memory based learning (MBL) models generally outperforming partial least squares regression and Cubist models. Our study suggests that while good-to-excellent results can be obtained without calibration transfer, for most of the cases presented in this study, PDS was necessary for unbiased predictions. The MBL models also outperformed the direct calibration models for most of the soil properties. For laboratories building new spectroscopy capacity utilizing existing spectral libraries, it appears necessary to develop calibration transfer using PDS or other calibration transfer techniques to obtain the least biased and most precise predictions of different soil properties.

## 1. Introduction

Recent research has increasingly focused on using diffuse reflectance spectroscopy (DRS) as a routine analysis tool to make useful predictions of numerous soil properties. Diffuse reflectance spectroscopy therefore provides the potential of offering a fast, cost-effective and non-destructive alternative to traditional soil laboratory analysis [1,2] using the reflectance of visible, near infrared and mid infrared wavelengths of light to predict soil properties [3]. In quantifying a soil property, DRS relies on multivariate and machine learning techniques to develop robust predictive models using the relationship between predictor variables (acquired spectra) and the response variable (a known soil property). Considerable effort has been placed on developing robust predictive models since it is meant to reduce model predictive error and can be deployed anywhere in space and time [4].

There are several conditions that must be met for successful deployment of predictive models to estimate the value of a given soil property for an unknown sample. First, an unknown sample should fall within the same domain (i.e., calibration space) used to develop the predictive model. Second, the spectral response due to the instrumental setup and environmental conditions impacts the reliability of the prediction [5,6,7]. While the first prerequisite (i.e., calibration space) can be addressed by building predictive models using soils representative of the broad geographic locations to which a model will be applied, the ideal solution for the second issue (i.e., instrumental and environmental differences) is to rescan every soil with the same spectrometer and under identical environmental conditions, which is practically not feasible. An alternative strategy to address the second issue is to use chemometric techniques to correct for instrumental and environmental differences [8]. Recent work also suggests that with certain combinations of spectrometers, spectral preprocessing and model choice, this second issue may have minimal impact on predictions [9].

A variety of research has focused on building large-scale soil spectral libraries at national, continental and global levels, with the purpose of supporting many activities relevant to soil analysis, proximal soil sensing, soil monitoring and digital soil mapping [10,11,12]. Generating large spectral libraries spanning a range of environmental, climatic, topographic and geographic conditions can be an important step forward in building robust predictive models to estimate properties of any new samples [13], and global initiatives such as the Global Soil Laboratory Network (GLOSOLAN), and Standards and Protocol for Measuring Soil Spectroscopy (IEEE) are beginning to tackle these issues. However, the spectral libraries are scanned using different spectrometers and often under different environmental conditions impeding the development of robust predictive models that would be successfully tested against new samples scanned under different instrumental and environmental conditions [6]. Additionally, direct use of predictive models built using these large spectral libraries on spectra acquired on a secondary spectrometers may result in unreliable predictions, unless a set of mathematical and statistical methods are used to adjust the spectra from the secondary spectrometer [14]. To account for the inherent differences in instrumental/environmental conditions, several standardization (i.e., calibration transfer) techniques have been developed.

These standardization techniques range from simple slope and bias correction (SBC) to complex multivariate regression approaches [15,16]. Among them, the most commonly used techniques in soil spectroscopy include SBC, direct standardization (DS), and piecewise direct standardization (PDS) [6,7]. In SBC, a simple correction factor can be used for adjusting the predicted values. This method generally improves predictions when the differences between the spectrometers are simple and systematic [17]. Direct standardization is a multivariate technique where a soil spectrum acquired on a secondary spectrometer is transferred to that of the primary spectrometer using a linear relationship described by the transformation matrix [18]. The limitation of this method is that the full spectrum of the secondary spectrometer is used to estimate the transfer matrix at each frequency [19]. As a result, there is increasing risk of model overfitting due to a limited number of transfer samples compared to the number of variables used to develop the transfer matrix [20,21]. Piecewise direct standardization addresses the potential overfitting problem associated with DS, by relating the signal intensity at each wavelength in the primary spectrometer to that within a small window of neighboring wavelengths acquired on the secondary spectrometer [22]. Since the coefficient vector for predicting the absorbance spectra at each wavelength is estimated using only a few wavelength channels in the secondary spectrometer, this method generally avoids the risk of overfitting [23,24]. A limitation of this method is that eigenvalue swapping between adjacent windows of the secondary spectrometer can result in discontinuities in the PDS transferred spectra. However, this limitation can be addressed by processing the spectra with derivatives and Fourier transformation prior to the deployment of PDS methods [15].

Although considerable effort has been placed on merging spectral libraries and developing robust predictive models using diffuse reflectance spectroscopy, it is still unclear whether predictive models built using soil samples scanned on a primary instrument can be directly applied to soil samples scanned on the secondary instrument. Here, we assess the potential of using predictive models developed on the large USDA National Soil Survey Center–Kellogg Soil Survey Laboratory (NSSC-KSSL) spectral library on spectra obtained on a secondary spectrometer in a different laboratory. Our overall objective was to assess the need for calibration transfer when using the NSSC-KSSL derived models on a secondary spectrometer. Specifically, we tested two different ways of applying the PDS calibration transfer approach in combination with three different modeling approaches. We hypothesized that spectra standardized using the PDS method can more accurately predict soil properties compared to untransformed spectra because the PDS method is able to deal with nonlinearities associated with instrumental and environmental effects by allowing for correction in the peaks and wavelength shifts in spectra.

## 2. Materials and Methods

Multiple methodological steps were necessary in order to address these two sets of research questions:Can useful predictions of soil properties be achieved on a secondary instrument without accounting for calibration transfer or having to build new models directly from the secondary spectra? Relatedly, how much of a loss in predictive performance is incurred by using spectra obtained on the secondary instrument?Can calibration transfer significantly improve predictions on the secondary instrument? What is the optimal combination of calibration transfer and predictive modeling approach?

To address the first set of questions, three modeling algorithms (partial least squares regression (PLSR), memory-based learning (MBL) approach and Cubist) were applied to the primary spectral library to predict several soil properties on two different sets of soil samples scanned on the primary and a secondary instrument, and one set of soil samples scanned only using the secondary instrument. New direct calibration models were also developed for each sample set using only the spectra from the secondary spectrometer and the predictive performance was evaluated using a leave-one-out cross validation approach.

To address the second set of questions, we developed a calibration transfer procedure using two different applications of a piecewise direct standardization (PDS) approach to the first sample set. The PDS-transformed secondary spectra from the second and third sample sets were then predicted using the three modeling algorithms and compared to the model performance without calibration transfer and to the model performance of the direct calibration results.

We focused our model assessment on five soil properties: three that are generally predicted very well from mid-infrared (MIR) spectroscopy—organic carbon (OC), calcium carbonate (CaCO_3_) and clay content; pH that has shown variable success and bulk density (BD) a property that is often difficult to predict accurately.

### 2.1. Soil Samples, Measured Properties and Scanning Spectrometers

Several sets of samples were used to assess the transferability of soil spectra from the primary spectrometer located at the Kellogg Soil Survey Laboratory and the secondary spectrometer located at the Woodwell Climate Research Center. The spectrometers and soil spectra at the Kellogg Soil Survey Laboratory and the Woodwell Climate Research Center are, hereafter, abbreviated as KSSL and Woodwell, respectively. The primary calibration or training set for model building consisted of a large fraction of the USDA National Soil Survey Center–Kellogg Soil Survey Laboratory (hereafter, referred to as NSSC-KSSL spectral library; *n* = 56,155) and the associated Soil Characterization Database.

A transfer set (Set I; *n* = 493) was selected using the Kennard–Stone algorithm [25] to represent the diversity in spectral responses seen in the USDA NSSC-KSSL spectral library (Table 1). Set I samples were scanned using KSSL and Woodwell spectrometers to serve two purposes: (1) to test the performance of the predictive models without calibration transfer and (2) to develop a PDS transfer matrix between the primary (KSSL) and secondary (Woodwell) spectrometer. These 493 samples were withheld when building the predictive models from the NSSC-KSSL spectral library.

Two prediction sets (Set II; *n* = 296 and Set III; *n* = 605) were used for independent performance assessment, and consisted of soil samples that have not been used in developing either the PDS transfer matrix or the predictive models. The Set II samples came from the mega-pit archived soils from across the National Ecological Observatory Network (NEON) in the USA scanned using the KSSL and Woodwell spectrometers. Analytical data for sample Sets I and II were determined at the KSSL laboratory using methodologies detailed elsewhere [26]. The Set III samples were from the land use/cover area frame statistical survey (LUCAS) of the European Union available through the European Soil Data Center (ESDAC), and scanned using the Woodwell spectrometer (Table 1). Toth et al. [27] provide details on laboratory methods for the Set III samples. Methods for determining CaCO_3_ and OC content were identical between the two laboratories. However, the method for determining clay content differed slightly between the KSSL and Set III samples with the KSSL using sedimentation only and Set III using a combination of sieving and sedimentation. Additionally, pH was measured using a 1:1 soil-to-water ratio by the KSSL laboratory but measured in a 1:5 suspension in the Set III dataset.

All KSSL spectra were acquired on a Bruker Vertex 70 FTIR (Bruker Optics, Billerica MA, USA) equipped with HTS-XT high throughput diffuse reflectance accessory (Bruker Optics, Billerica MA, USA). The Woodwell samples were also scanned on a Bruker Vertex 70 FTIR but with a Pike AutoDiff (Pike Technologies, Fitchburg WI, USA) diffuse reflectance accessory. In all cases, spectra were acquired on dried and finely milled soil samples. Details of operating conditions can be found in Dangal et al. [13].

### 2.2. Spectral Preprocessing and Calibration Transfer

Spectra acquired using KSSL and Woodwell spectrometers have different spectral ranges and resolution. Therefore, all spectra were truncated to a common spectral range of 4000–628 cm^−1^, and interpolated to a resolution of 2 cm^−1^. The interpolation was carried out using the prospectr package in R (R statistical computing).

We then developed PDS transfer functions using both the raw and first derivative transformed spectra. While PDS is often applied to raw spectra, the application of PDS to raw spectra can introduce spectral noise due to abrupt change in the shape of the model leading to discontinuities in the spectra; therefore, spectral preprocessing prior to building the transfer matrix can help to reduce noise and discontinuities in the spectra [28]. As a result, we developed two different PDS transfer functions using either the raw or first derivative spectra to examine whether spectral preprocessing prior to PDS can improve predictions. We chose the first derivative because it helps to reduce baseline offset and extracts useful information related to the prediction of response variables [29,30]. After the development of PDS transfer functions, regions showing atmospheric CO_2_ features (2389–2268 cm^−1^) were removed.

Whether working with raw or first derivative transformed spectra, three parameters are required in developing PDS transfer functions: (1) window size; (2) number of principal components (PC) and (3) number of samples in the transfer set. We used the three data point window to develop PDS transfer functions because increasing the window size decreased the model reliability by introducing shifts in spectral peaks. The number of PC to retain for each spectral window was set to one because increasing the number of PC introduced large spectral noise (Appendix A). However, the number of PC was not responsible for introducing shifts in spectral peaks. Since the PDS operates within a moving window introducing edge effects due to a lack of sufficient data points to form a complete window, the data points that are unable to form a complete window are either removed or interpolated using some statistical techniques [31]. In this study, we removed the data points at the edge that are unable to form a complete window. The third parameter for successful PDS transfer requires optimization of the number of transfer samples and the selection of these samples from the large library [32]. We limited the number of samples to approximately 1% of the NSSC-KSSL spectral library and selected these samples using the KS algorithm to pick the most representative samples in the PC space (Figure 1a).

### 2.3. Outlier Detection and Predictive Model Development

Outlier detection follows a two-step procedure. In the first step, we detected outliers from the calibration set scanned using the KSSL spectrometer by building PLSR models and testing their performance against all samples in the spectral library. We optimized outlier detection using a standard deviation threshold (0.1–3 at increments of 0.02) from the 1 to 1 line to pick up the maximum of 1% of the samples as outliers [13]. To accomplish this, we picked samples that are farther away from the optimized standard deviation threshold. The original NSSC-KSSL library minus the transfer set (*n* = 493) consisted of 56,155 samples. Using the standard deviation threshold, we retained 55,598 from the NSSC-KSSL library to build the predictive models of OC (Appendix A). For other soil properties, we performed outlier screening using the standard deviation threshold to retain 13,667, 9365, 37,187 and 39,347 samples before building the final predictive models of BD, CaCO_3_, clay and pH, respectively. 

In the second step of outlier detection, we used the *F*-ratio criteria to identify samples with spectra that fall outside the calibration space and flagged the predictions originating from these spectra as potentially unreliable for independent validation sets II and III (Appendix A). The *F*-ratio approach relies on developing a probability distribution of the spectra to flag any samples with a probability greater than 0.99 as outliers [13]. It is important to recognize that outliers using the *F*-ratio are detected by developing a probability distribution function of the spectra scanned using the same spectrometer (i.e., primary spectra) for set I and set II samples. For set II samples, we detected outliers on KSSL, Woodwell and Woodwell_PDS_ spectra, while for set III samples, we detected outliers on Woodwell and Woodwell_PDS_ spectra. The outlier detection was done for both baseline corrected and first derivative spectra.

The predictive models of BD (g/cm^3^), CaCO_3_ (%), clay (%), OC (%) and pH were developed using two approaches. In the first approach, we used NSSC-KSSL spectral library acquired on the primary spectrometer (KSSL; Appendix A) and tested against the following combination of spectra: (1) spectra acquired on the primary spectrometer (KSSL); (2) spectra acquired on the secondary spectrometer (Woodwell) and (3) PDS transformed secondary spectra (Woodwell_PDS_). The predictive models were built using two preprocessing techniques: a simple baseline correction and more complex first derivative transformation. The spectra were baseline corrected by subtracting spectral responses corresponding to each sample from their minimum value. For the first derivative transformation, we used a finite difference algorithm with a lag size of one. In the second approach, we developed the predictive model using spectra acquired on the secondary Woodwell spectrometer (hereafter, referred to as direct calibration model). The purpose of the direct calibration is to examine whether a large primary (KSSL) spectral library applied on soil spectra scanned using a secondary (Woodwell) spectrometer can perform better compared to the models on soil spectra scanned using the secondary (Woodwell) spectrometer. Again, both baseline corrected and first derivative spectra were tested in combination with the same modeling algorithms to develop the direct calibration models on Woodwell spectra from set I, set II and set III samples. Direct calibration model performance was assessed using the leave-one-out cross validation.

Each of the above options were tested using three different modeling approaches: one multivariate regression (PLSR) and two machine learning models (MBL and Cubist) [33,34,35]. The PLSR is a linear multivariate model that reduces the number of predictor variables into few latent variables to maximize the covariance between the predictor and response variables [33]. The MBL is a local modeling approach, where a function (i.e., PLSR, weighted average PLSR or Gaussian process regression—GPR) is developed for each sample in the prediction set using a set of spectrally similar neighbors as defined by different similarity search methods [36]. Cubist uses a classification and regression tree (CART) algorithm to construct trees, followed by the development of a set of multivariate models associated with a set of rules at each terminal node [34]. Details of the mathematical formulation used to develop MBL, Cubist and PLSR models are available in Dangal et al. [13].

In the case of the PLSR models, the number of latent variables was optimized using the one-sigma approach after performing a 10 segment cross validation for all soil properties. In this approach, the model with fewer PC components that falls within one standard error compared to the overall best model is retained as the optimal PC using randomized segments. For the MBL models, we used weighted average PLSR to fit a local function for each target sample. The number of samples to fit a target function was selected in PC space using a distance threshold defined by their minimum (0.4) and maximum (5.0) values. Within each distance limit, the model finds the spectral neighbors and uses those neighbors to fit a function relevant to the target sample. If the total number of samples selected at each threshold distance is smaller or larger than the user defined minimum and maximum number of neighbors, the function will be forced to select the specified minimum or maximum number of samples, respectively. In this study, we limited the minimum and maximum number of samples to 100 and 200, respectively. The selection of the minimum and maximum number of samples is arbitrary, and intended to limit the number of neighbors across each distance threshold to find the optimized model. In the case of Cubist, the number of committees that defines the boosting iterations was set to one, and the maximum number of rules was set to 100.

### 2.4. Model Performance

To assess the goodness of fit of the predictive models and their performance against independent validation sets, we used the coefficient of determination (R^2^), ratio of performance to the interquartile distance (RPIQ), root-mean-square error (RMSE) and model bias. Details on the estimation of R^2^, RMSE and model bias are available in Dangal et al. [13]. Here, we defined an additional criteria based on R^2^: R^2^ ≥ 0.85 (excellent models), 0.75 ≤ R^2^ < 0.85 (good models), 0.65 ≤ R^2^ < 0.75 (fair models) and R^2^ < 0.65 (non-reliable models). Additionally, RPIQ measures the ratio of the interquartile distance (Q3-Q1) to RMSE and better represent the spread of the population [37]. Since RPIQ scales the spread of the data by RMSE, it allows for the comparison of model performance across validation sets and soil properties. Assessment of the model performance using the above criteria is arbitrary and has been used in this study to rank the prediction performance of a range of soil properties. We also compared model performance with and without PDS for all soil properties using the percent difference in RMSE between the primary and secondary spectra. In the case where MIR scans using the primary spectrometer were unavailable (Set III), we calculated the percent difference in RMSE by comparing the predictions using the PDS spectra against the spectra without PDS.

All data processing and analysis were performed in R statistical software [38]. The PDS calibration transfer routine was developed by using the pls package. Spectral preprocessing, development of PDS transfer function and predictive models, and the evaluation of model performance for a range of soil properties were done using the Kubernetes cluster available through cloudops (https://www.cloudops.com/). The Kubernetes cluster consists of 50 vCPUs and 240 GB of memory partitioned into a master node and six worker nodes. The cluster is built using Ubuntu, an open source Linux distribution system and R 3.6.1 version. Total computational time was approximately 130 CPU hours. The script for developing PDS transfer, predictive models and prediction performance analysis is freely available in the Woodwell Climate Research Center GitHub (https://github.com/whrc).

## 3. Results

### 3.1. Performance of KSSL Predictive Models

The PLSR and Cubist produced good to excellent calibration models of all soil properties (R^2^ ≥ 0.81) using samples from the NSSC-KSSL library on baseline corrected and first derivative spectra (Appendix A). The MBL does not produce a predictive model since a new model is built specific to each new sample predicted. Cubist, regardless of preprocessing, outperformed the PLSR calibration models for these same properties. Slightly better models were produced with Cubist using the baseline corrected spectra for all the properties expect OC. For OC, the predictive models produced using Cubist were similar regardless of spectral preprocessing (R^2^ = 0.99, RMSE = 0.90).

### 3.2. Performance of KSSL Predictive Models on KSSL and Woodwell Spectra

When the NSSC-KSSL spectral library was used to develop modeling algorithms to test the performance on set I samples, the MBL demonstrated the best performance (higher RPIQ) on both the primary (KSSL) and secondary PDS (Woodwell_PDS_) spectra (Appendix A). Comparison of the R^2^ values showed that all three modeling approaches demonstrated good to excellent performance (R^2^ ≥ 0.75) on the primary (KSSL) spectra for all soil properties except BD. On the secondary (Woodwell) spectra, both the PLSR and MBL showed good to excellent performance of all soil properties except BD, while the Cubist showed large increases in RMSE when moving from the primary spectra (KSSL) to the Woodwell spectra except for OC (Appendix A). Spectral preprocessing showed different results with MBL built on first derivative spectra outperforming baseline corrected spectra, while Cubist built on baseline corrected spectra outperforming first derivative spectra for all soil properties. The PLSR, on the other hand, showed mixed results with better performance of CaCO_3_, OC and pH on the first derivative spectra.

On the independent validation set II consisting of primary spectra, the RPIQ values across different models demonstrated that MBL tested on the primary (KSSL) spectra outperformed the PLSR and Cubist for all soil properties except pH (Appendix A). The R^2^ values for all three modeling approaches showed good to excellent performance with R^2^ ≥ 0.86 for clay, OC and pH. For these properties, all three modeling approaches produced a better fit (higher R^2^) compared to the performance using the transfer set (Set I), although higher RPIQ values were obtained using set I samples. When the secondary (Woodwell) spectra were used, both the PLSR and MBL still showed good to excellent performance (R^2^ ≥ 0.75); however, there was a drop in performance when moving from primary (KSSL) to secondary (Woodwell) spectra. The R^2^ of the MBL dropped by 0.03, 0.15 and 0.18, while that of the PLSR dropped by 0.02, 0.07 and 0.01 for OC, clay and pH; respectively, when the KSSL predictive models were tested on Woodwell spectra. The Cubist model, on the other hand, showed unreliable performance on the Woodwell spectra for all soil properties except OC. First derivative spectral preprocessing produced slight better results on both primary (KSSL) and secondary (Woodwell) spectra for all soil properties (except pH and clay on Woodwell spectra using MBL and PLSR, respectively). Similar to set I results, the Cubist model built on baseline corrected spectra outperformed first derivative spectra for all soil properties.

When the three modeling algorithms developed on the NSSC-KSSL spectral library were tested on Set III samples of European origin, comparison of the RPIQ values showed that the MBL outperformed the PLSR and Cubist for all soil properties expect clay predicted using the baseline offset spectra (Appendix A). The PLSR and MBL showed excellent performance (R^2^ ≥ 0.85) of CaCO_3_ and OC on Woodwell spectra, while fair performance (0.65 ≤ R^2^ < 0.75) of pH, and mixed results for clay. Since, set III samples were only scanned using the Woodwell spectrometer, we could not assess the model performance using the KSSL spectra. Similar to results from set I and set II samples, Cubist showed unreliable performance on Woodwell spectra. First derivative spectra produced better results using the PLSR and MBL models for all soil properties except CaCO_3_ (using MBL) and clay (using PLSR).

### 3.3. The Effect of Calibration Transfer on Secondary (Woodwell) Spectra

Before calibration transfer, there were large differences between the raw spectra of the same soil samples acquired on the two spectrometers (Figure 1a–c) with 76.5% of the variance in spectra explained by the overall differences in the spectrometer due to baseline shift by the first PC. After a first derivative transformation, the differences in the spectrometer due to baseline shift is removed, with the first and second latent variable explaining 45.7% and 12.7% of the variance in spectra (Figure 1d–f). As a result, both the KSSL and Woodwell spectrometer displayed similar characteristics in PC space. The PDS procedure successfully eliminated most of the differences between the primary and secondary raw spectra (Figure 1). However, closer inspection of Figure 1a revealed that because there was more diversity in spectral characteristics (i.e., greater spread in PCA space) of the KSSL Set I samples than the Woodwell Set I samples, the PDS transformed Woodwell Set I samples (Woodwell_PDS_) could not reproduce this full range of spectral response as seen in the KSSL spectra.

### 3.4. Performance of KSSL Predictive Models on Calibration Transfer (Woodwell_PDS_) Spectra

When the KSSL predictive models were tested on PDS transfer (Woodwell_PDS_) spectra consisting of Set I (transfer set) samples, the MBL showed the best model performance with higher RPIQ for all soil properties. Comparison of the R^2^ values also showed good to excellent performance of all soil properties except BD, while the Cubist and PLSR showed good to excellent performance of CaCO_3_, clay and OC and fair performance of pH (Appendix A). Among all three modeling approaches, Cubist experienced the greatest drop in performance using the Woodwell_PDS_, with R^2^ dropping by 0.05, 0.18, 0.23 and 0.28 for CaCO_3_, clay, pH and BD; respectively, compared to the performance using the KSSL spectra. The MBL tested on Woodwell_PDS_ spectra decreased RMSE by 7%, 13%, 20%, 25% and 31% for BD, CaCO_3_, pH, OC and clay; respectively, compared to the Woodwell spectra (Figure 2). Likewise, the PLSR improved RMSE by 5-9% for all soil properties except pH.

On the independent set II samples, with the exception of OC on baseline-offset spectra, the MBL was the best model (higher RPIQ) on both baseline offset and first derivative PDS (Woodwell_PDS_) spectra for all soil properties. Comparison of the PLSR and MBL showed good to excellent performance when tested on Woodwell_PDS_ spectra for all soil properties except BD (Appendix A). With the exception of OC, Cubist demonstrated inconsistent performance ranging from fair to unreliable predictions of different soil properties. Calibration transfer improved MBL model performance relative to Woodwell spectra with RMSE decreasing by 13%, 15%, 23% and 31% for BD, clay, pH and OC, respectively (Figure 3). Similar results were found when using PLSR models. Model RMSE decreased by 7%, 7% and 33% for BD, pH and OC, respectively, for the Woodwell_PDS_ compared to Woodwell spectra. With the exception of BD, both the PLSR and MBL produced slightly better performance on the Woodwell_PDS_ spectra developed using a first derivative PDS transfer matrix.

In the final set (Set III) of European origin, comparison of the RPIQ values across different models on Woodwell_PDS_ spectra showed that MBL consistently outperformed other models for all soil properties on the first derivative PDS (Woodwell_PDS_) spectra. On the baseline-offset spectra, comparison of the RPIQ showed that MBL outperformed PLSR and Cubist for CaCO3, OC and pH, while slightly better performance of clay were obtained using the PLSR. When R^2^ was used for model comparison, the PLSR and MBL showed good to excellent performance of all soil properties after PDS transformation (except pH using the PLSR model). Similar to results from set I and set II, the Cubist showed fair to unreliable predictions when tested using the Woodwell_PDS_ spectra for all soil properties except OC. Compared to the performance using the Woodwell spectra, the MBL tested on Woodwell_PDS_ spectra decreased RMSE by 3%, 28% and 36% for clay, pH and OC (Figure 4). For CaCO_3_, however, the MBL on Woodwell spectra outperformed Woodwell_PDS_ spectra by 12%. The RMSE of PLSR model for OC decreased by 17% using the Woodwell_PDS_ spectra compared to the Woodwell spectra, but for other soil properties the performance were not different between the Woodwell and Woodwell_PDS_ spectra. For both the PLSR and MBL models, the Woodwell_PDS_ spectra obtained using the raw PDS transfer model produced slightly better predictions of all soil properties (except CaCO_3_) compared to the Woodwell spectra.

### 3.5. Performance of Woodwell Direct Calibration Models Using Leave-One-Out Cross Validation

Comparison of the predictions of soil properties using the NSSC-KSSL spectral library against the direct calibration model is confounded a bit by the fact that the performance of the models were tested using different approaches (i.e., leave-one-out cross validation in the case of direct calibration versus separate prediction set in the case of NSSC-KSSL library). With the exception of BD and CaCO_3_, the MBL built using NSSC-KSSL spectral library demonstrated better predictions on KSSL and Woodwell_PDS_ spectra compared to the direct calibration. However, with the set II samples, the direct calibration models were similar to (i.e., BD and clay) or slightly better (i.e., OC and pH) than the Woodwell_PDS_ spectra (Figure 3). In case of set III samples, the results were similar to set I for CaCO_3_ and clay, with results from direct calibration similar to Woodwell_PDS_ spectra (Figure 4). However, direct calibration outperformed Woodwell_PDS_ spectra in the case of pH and vice versa in the case of OC. 

## 4. Discussion

Diffuse reflectance spectroscopy (DRS) is emerging as a viable alternative for rapid and accurate characterization of soil properties compared to time-consuming traditional soil laboratory analysis. As a result, multiple spectral databases consisting of soil samples scanned using different spectrometers are available to produce accurate predictions of soil properties [10,11,39]. However, there is debate as to whether a predictive models developed using spectra scanned on a primary spectrometer can be directly applied to spectra scanned on a secondary spectrometer or if there is a need for spectra standardization [6,15,28]. Using the large NSSC-KSSL spectral library with and without PDS, we demonstrated that (1) fair to good predictions of different soil properties could be achieved without accounting for calibration transfer, (2) spectral preprocessing with first derivative transformation of secondary spectra slightly improved overall model performance compared to the baseline corrected spectra, (3) calibration transfer using the PDS on the secondary spectra generally improved prediction of soil properties by reducing the model bias and improving the RMSE, (4) across soil properties, the MBL was most often the best performing model on PDS spectra in the independent prediction sets and (5), in many cases, use of the MBL on the PDS transformed secondary spectra outperformed direct calibration models built from the secondary spectra.

### 4.1. Model Performance without Accounting for Calibration Transfer

Our results demonstrated that model performance on soil spectra from the secondary spectrometer (Woodwell) was largely dependent on the type of predictive model and the spectral preprocessing [6,40]. Both the PLSR and MBL approaches developed using the NSSC-KSSL spectral library acquired on the primary spectrometer (KSSL) demonstrated fair to excellent prediction of most of the soil properties on the secondary (Woodwell) spectra, while the Cubist showed the greatest drop in model performance except for OC. It is important to recognize that the nature of the target function developed using different methods has a strong influence on model performance as these functions determine the importance of different wavelength bands during predictions [41]. Both the PLSR and MBL uses similar techniques while fitting a target function [13,42], but Cubist relies on the use of rules to subset the data, where a linear model is fitted along each terminal nodes [34]. Although Cubist produced best results when tested on spectra acquired using the same spectrometer (KSSL), the drop in model performance on Woodwell spectra is likely due to the complexity of the relationship between the response and predictor variables. With complex models like Cubist, there is an increasing risk of model overfitting, where the accuracy of the training set is high, but the models cannot be successfully applied to new samples [43]. In the case of set III samples, greater model errors were obtained when predicting clay and pH, which is at least partially the result of different underlying analytical data. Validation data for the Set III came from a European laboratory applying slightly different methods for clay and pH and have different ranges of the properties.

Our results also demonstrate that spectral preprocessing with first derivative transformation can slightly improve predictions compared to baseline corrected spectra. Spectral preprocessing is generally employed to minimize the contribution of spectral features not related to the prediction of a given soil property or that cannot be handled by the modeling techniques [44]. While studies have shown that spectral preprocessing improves the performance of predictive models [2,45], our results demonstrate that spectral preprocessing can reduce some of the additional variability introduced by scanning the soils on a secondary spectrometer (Appendix A).

### 4.2. Effect of Calibration Transfer on Spectra Acquired Using Different Spectrometers

Application of PDS models developed using KSSL as the primary and Woodwell as a secondary spectrometer demonstrated that the differences in the scanning spectrometers and/or environments were reduced when the PDS technique was applied to both the raw and first derivative spectra scanned using the secondary (Woodwell) spectrometer. This was largely because the PDS reduced variations in intensity and wavelength shifts through the development of local multivariate models across small wavelength regions [15,28]. For the transfer set soils, large differences in the spectral responses existed between the primary and secondary spectra mostly associated with differences in instrumental baselines when using the raw spectra (Figure 1a–c). When these raw spectra were used to develop a transfer matrix using the PDS, the differences in the mean spectral responses between the KSSL and Woodwell spectra were reduced for both transfer (set I) and independent prediction (set II) sets. However, the best results (i.e., smallest difference between primary and PDS-transformed secondary spectra) were obtained when the PDS transfer matrix developed using the first derivative spectra (Appendix A). This is likely because application of PDS on raw spectra can introduce spectral noise due to abrupt change in the shape of the model leading to discontinuities in the spectra [46]. Spectral preprocessing of primary and secondary spectra using derivatives and/or Fourier transformation prior to the PDS transfer can remove noise by reducing baseline offset, compensating instrumental drift and extracting information related to the prediction of the response variables [28]. While spectral preprocessing using first derivative spectra produced a slightly better PDS transfer model, predictions of most of the soil properties were only marginally better compared to predictions from the raw PDS transfer model (Figure 2, Figure 3 and Figure 4).

### 4.3. The Need for Calibration Transfer

Using the NSSC-KSSL spectral library as a primary spectrometer, calibration transfer improved predictive model performance of all soil properties when tested against the transfer samples (Set I). These models also improved performance for most of the soil properties on truly independent prediction sets consisting of spectra from the USA (Set II) and European (Set III) origin. Our findings are consistent with several studies that show improved model performance after PDS [7]. Improvement in model performance after the PDS can be attributed to two main reasons. First, the transfer set used to develop the transfer function should be representative of the large spectral library [47], and second, the predictive models should contain sufficient samples to represent the distribution and variability of soil where the model is to be applied [48]. When mean correlation of all samples among the primary (KSSL), secondary (Woodwell) and PDS (Woodwell_PDS_) spectra were used to measure spectral variations, PDS decreased the spectral variability of both the transfer (KSSL) and independent set II datasets (Appendix A). In addition, the variations in soil properties corresponding to the transfer set was large enough to fully capture the variations in the soil properties of the independent set II and set III datasets. As a result, the PDS method produced better results compared to the untransformed spectra.

For some properties such as clay and CaCO_3_, deployment of calibration transfer on the truly independent sets did not improve the performance even when the NSSC-KSSL library was used as a primary spectrometer. This is possibly due to the fact that the diversity of these soil properties were not well represented in either the transfer or the calibration set used to develop the predictive model [6,49,50]. For example, Ge et al. [6] found improved performance of OC with PDS transfer spectra, but suggested that other soil properties might exhibit different variability in principal component space that could introduce differences in the prediction performance of the soil property under consideration with and without the calibration transfer. We demonstrated here that, against independent validation sets, the PDS technique was able to improve predictions for 6 out of 8 properties, regardless of whether the PDS transfer function was built on raw or first derivative spectra.

It is important to recognize that although calibration transfer on set II and set III data demonstrated the best predictions, performance of the predictive model on the secondary spectra (Woodwell) were generally really good and not that different for OC, CaCO_3_, clay and pH suggesting that PDS might not be necessary for satisfactory spectroscopy-based estimates of these properties. The first derivative transformed secondary (Woodwell) spectra outperformed the baseline corrected spectra for almost all soil properties across all three sample sets. In the PC space, both the baseline corrected and first derivative spectra picked up similar number of samples as outliers when the *F*-ratio approach was used to detect spectral outliers (Appendix A). With the Woodwell_PDS_ spectra on both the set II and set III samples, the first derivative transformation picked up a slightly higher number of samples as outliers compared to the baseline corrected spectra (Appendix A). However, the secondary Woodwell spectra performed reasonably well indicating that models built using a large spectral library can still reliably predict a target variable outside of their broad distribution range.

### 4.4. Direct Calibration Using WOODWELL Spectra Are Not Always the Best Model

Comparison of the prediction statistics from the direct calibration using Woodwell spectra to the application of the KSSL predictive models to the Woodwell_PDS_ spectra suggests that for many soil properties there is little if any gain in predictive performance through direct calibration. The performance of the direct calibration models built using Woodwell spectra are comparable to that of the KSSL predictive models tested on Woodwell_PDS_ spectra for all three sets. In other words, despite all the potential additional variance introduced by use of a secondary spectrometer and PDS transformation, the large NSSC-KSSL library-based models often showed superior performance. Although some studies have shown that prediction using the PDS transfer spectra can be better than those of a direct calibration [28], others have indicated that the direct calibration generally outperforms the prediction using the PDS transfer spectra [50]. This could be attributed to the quality of the spectrometer with the PDS outperforming a direct calibration when the signal on one spectrometer is standardized to that of a higher quality spectrometer. Additionally, most of these studies used the same training set for direct calibration to develop the calibration model using the spectra scanned on the same spectrometer. However, in this study, we used a much larger training set consisting of samples from the USA, while the direct calibration were limited to the number of samples in the independent prediction sets. It is important to point out that the results from using the Woodwell_PDS_ spectra are not directly comparable with the results from the direct calibration model because of different methods (i.e., leave-one-out cross validation vs. the separate prediction set) used for evaluating model performance. The comparison is done to provide a conservative estimate of the loss/gain in predictive power between the primary to secondary spectra model vs. the models developed using spectra scanned on the same instrument.

### 4.5. Best Model Performance

Overall, consistent to our earlier work [13], these results indicate that the local modeling approach (MBL) only slightly outperformed both Cubist and PLSR models when tested against soil spectra acquired using the same spectrometer used to develop the predictive models. However, when confronted with spectra acquired on different spectrometers, relative model performance deviated substantially with Cubist providing unreliable predictions even after calibration transfer. On the other hand, the MBL and PLSR tested on secondary and transfer spectra developed using a PDS transfer model only showed a slight drop in performance relative to predictions from the primary spectra. Although MBL was clearly a better approach when tested on independent sets consisting of spectra from the same spectrometer, the differences in the choice of model when tested on soil spectra acquired using different spectrometer might be associated with the way both PLSR and MBL handle new spectra added to the library. For example, the PLSR is less influenced by the location of the samples in the PC space because a global model is used to predict the target sample, while the MBL is more sensitive to the location of prediction samples in PC space. Any small changes in the location of samples can introduce large prediction errors because the neighboring samples selected in PC space can be quite different than the spectral features of the transfer spectra used to make prediction of a given soil property [42]. The mixed results of whether or not PDS was necessary and which combination of spectral pretreatment and model form (PLSR or MBL) produced the best predictions clearly indicates that there is unlikely to be a one-size-fits-all solution to the widespread use of large spectral libraries in multiple laboratories each collecting spectra on different spectrometers.

## 5. Conclusions

Using the large NSSC-KSSL mid-infrared spectral library applied to primary soil spectra (i.e., acquired on same instrument) and secondary soil spectra (i.e., acquired on a different instrument) from two distinct geographical regions (the USA and Europe), we evaluated the model performance with and without calibration transfer. Our results show that predictions were generally quite good without calibration transfer. However, in many cases, calibration transfer using PDS improved the performance of a target soil sample scanned on the secondary spectrometer by reducing the bias and improving model fit (higher R^2^). These results are particularly promising given the growing need to develop cost-effective methods for estimating soil properties by combining multiple spectral libraries scanned using different spectrometers at the local, regional and global scales.

The fact that across soil properties and sample sets, the best performance was achieved with different combinations of transformation, PDS application and model choice suggests that more research is necessary to work out optimal transfer approaches. Having to distribute limited number of samples to each laboratory that wants to start using a large spectral library is not a sustainable solution. Sample-free [51] and single standard approaches [52] need to be evaluated for their potential in mid-infrared spectroscopy. Additionally, ensemble-learning approaches [53] already common in many earth system science and pedometric applications can overcome the need to choose a single optimal model. Given the rapid rise in interest in diffuse reflectance spectroscopy across multiple disciplines and new international efforts to standardize soil spectroscopy methods, the future of soil spectroscopy is bright.

## Figures and Tables

**Figure 1 sensors-20-06729-f001:**
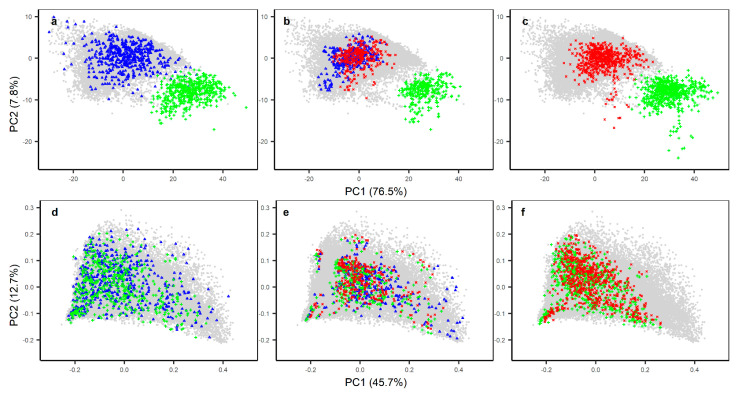
Relative locations of the transfer set (**a**,**d**) and two independent validation sets (**b**,**c**,**e**,**f**) in principal component space compared to the USDA National Soil Survey Center–Kellogg Soil Survey Laboratory (NSSC-KSSL) mid-infrared (MIR) library (grey dots) using the raw (top panel) and first derivative (bottom panel) spectra. The blue triangles are samples scanned using the primary KSSL spectrometer. The green plus signs are the samples scanned on the secondary Woodwell spectrometer and the red crosses are the Woodwell spectra after application of a piecewise direct standardization (PDS) transfer matrix built on the raw spectra using the transfer set.

**Figure 2 sensors-20-06729-f002:**
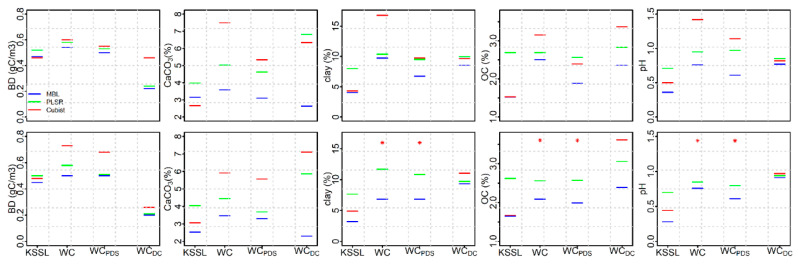
Root mean square error (RMSE) of KSSL, Woodwell (WC), Woodwell_PDS_ (WC_PDS_) and Woodwell_DC_ (WC_DC_) spectra representing transfer set (set I) for all three modeling approaches (memory-based learning (MBL), partial least squares regression (PLSR) and Cubist). The top and bottom panel shows the RMSE estimated using the prediction from the baseline corrected and first derivative spectra, respectively. The Woodwell_DC_ are the Woodwell spectra predicted using the direct calibration and leave-one-out cross validation. The red asterisk symbol shows that the RMSE using Cubist are too high to be included within the *y*-axis range for the given soil property.

**Figure 3 sensors-20-06729-f003:**
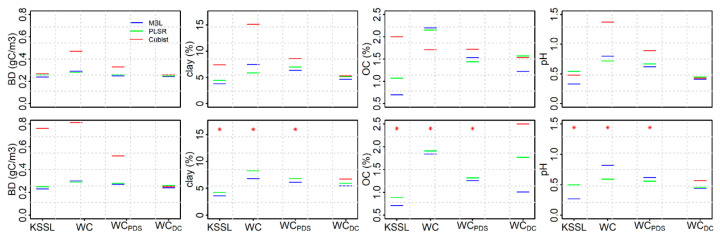
Root mean square error (RMSE) of KSSL, Woodwell (WC), Woodwell_PDS_ (WC_PDS_) and Woodwell_DC_ (WC_DC_) spectra representing independent validation set of the US origin (set II) for all three modeling approaches (MBL, PLSR and Cubist). The top and bottom panel shows the RMSE estimated using the prediction from the baseline corrected and first derivative spectra, respectively. The Woodwell_DC_ (WC_DC_) are the Woodwell (WC) spectra predicted using the direct calibration and leave-one-out cross validation. The red asterisk symbol shows that the RMSE using Cubist are too high to be included within the *y*-axis range for the given soil property.

**Figure 4 sensors-20-06729-f004:**
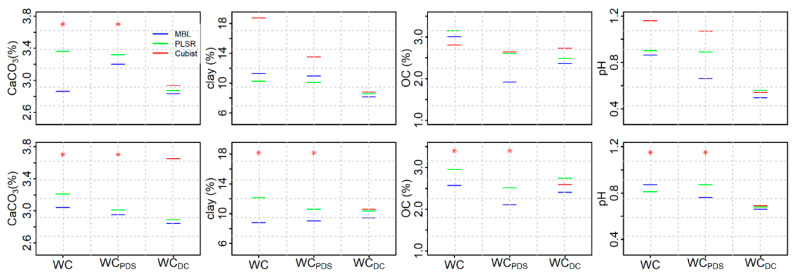
Root mean square error (RMSE) of Woodwell (WC), Woodwell_PDS_ (WC_PDS_) and Woodwell_DC_ (WC_DC_) spectra representing independent validation set of the European origin (set III) for all three modeling approaches (MBL, PLSR and Cubist). The top and bottom panel shows the RMSE estimated using the prediction from the baseline corrected and first derivative spectra, respectively. The Woodwell_DC_ (WC_DC_) are the Woodwell (WC) spectra predicted using the direct calibration and leave-one-out cross validation. The red asterisk symbol shows that the RMSE using Cubist are too high to be included within the *y*-axis range for the given soil property.

**Table 1 sensors-20-06729-t001:** Summary statistics of training set used to apply all three modeling algorithms, transfer sets (I) used to develop calibration transfer and independent prediction sets (II and III) used for predicting different soil properties.

Samples	Variables	*N*	Mean	Median	SD	Q25	Q75
Training Set	BD (g/cm^3^)	13,667	1.15	1.26	0.44	0.95	1.46
	CaCO_3_ (wt %)	9365	10.01	4.92	12.88	1.04	14.72
	clay (wt %)	37,187	22.71	20.93	15.79	3.15	5.70
	OC (wt %)	55,598	8.30	1.33	14.86	0.42	5.39
	pH	39,347	6.41	6.26	1.29	5.41	7.54
Set I	BD (g/cm^3^)	110	1.76	1.78	0.27	1.61	1.92
	CaCO_3_ (wt %)	216	12.39	3.33	18.53	0.32	15.53
	clay (wt %)	321	24.58	20.67	19.54	7.41	37.22
	OC (wt %)	420	6.68	0.81	12.85	0.24	5.60
	pH	341	6.58	6.50	1.48	5.34	7.93
Set II	BD (g/cm^3^)	290	1.23	1.31	0.42	1.04	1.52
(NEON)	clay (wt %)	286	17.21	13.00	14.22	5.63	25.98
	OC (%)	296	3.64	0.48	9.34	0.13	2.04
	pH	296	6.40	5.84	1.41	5.34	7.90
Set III	CaCO_3_ (wt %)	605	4.66	0.10	10.45	0.00	2.40
(LUCAS)	clay (wt %)	605	21.68	17.00	17.75	7.00	34.00
	OC (wt %)	605	5.14	1.99	9.66	1.14	3.45
	pH	605	6.32	6.32	1.31	5.21	7.53

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
