# Peer review of "Is Standardization Necessary for Sharing of a Large Mid-Infrared Soil Spectral Library?"

_sensors, 2020, doi:10.3390/s20236729_

Round 1

Reviewer 1 Report

In this manuscript the authors evaluated various spectral datasets from soil samples to ascertain the need for spectral standardization. This study is a good first step towards the application of diffuse reflectance spectroscopy in analyzing vast amount of soil samples given the growing need to develop cost effective methods. These vast data sets across different continents can be analyzed by using clever machine learning approaches.

Author Response

Comments: In this manuscript the authors evaluated various spectral datasets from soil samples to ascertain the need for spectral standardization. This study is a good first step towards the application of diffuse reflectance spectroscopy in analyzing vast amount of soil samples given the growing need to develop cost effective methods. These vast data sets across different continents can be analyzed by using clever machine learning approaches.

Response: We thank the reviewer for the comments. We did not see any comments from the reviewer that need to be addressed. We have revised the introduction and research design to improve the flow of manuscript.

Reviewer 2 Report

This work presented the assessment of using USDA predictive models to identify correctness utilizing existing spectral libraries. All data showed in the manuscript were sufficient and the explanations seemed reasonable. The subject matter is appropriate and the quality of the presentation is adequate. Before it can be recommended to accept for publication in the Journal, a revision is needed addressed:

  1. In conclusions section, there are unclear descriptions, rearrangement is necessary.
  2. Important methodological developments should certainly present in the manuscript, but some routine analysis methods or procedures needn’t show in “materials and methods” section, just marked with links/references or directly move to SI section.
  3. Although the manuscript title is short and simply, however it was difficult to understand clearly and directly. Suggest the title rewriting in order to reflect the text and meaning of the manuscript.

Minor:

  1. In materials and methods section, all subtitles should be renumbered. For instance, Line 155 should be changed as “2.2 Spectral preprocessing and calibration transfer”, Line 182 “2.3 …”, …, and etc.
  2. Important literatures are cited, some of them were too old in your 53 references.

Author Response

This work presented the assessment of using USDA predictive models to identify correctness utilizing existing spectral libraries. All data showed in the manuscript were sufficient and the explanations seemed reasonable. The subject matter is appropriate and the quality of the presentation is adequate. Before it can be recommended to accept for publication in the Journal, a revision is needed addressed:

Comment 1: In conclusions section, there are unclear descriptions, rearrangement is necessary.

Response: We thank the reviewer for the response. We have revised the conclusion section (particularly the first paragraph) to clarify that although good predictions were possible without calibration transfer, PDS was necessary to get unbiased predictions.

Using the large NSSC-KSSL mid-infrared spectral library applied to primary soil spectra  (i.e. acquired on same instrument) and secondary soil spectra (i.e. acquired on a different instrument) from two distinct geographical regions (the USA and Europe), we evaluated the model performance with and without calibration transfer. Our results show that predictions were generally quite good without calibration transfer. However, in many cases, calibration transfer using PDS improved the performance of a target soil sample scanned on the secondary spectrometer by reducing the bias and improving model fit (higher R2). These results are particularly promising given the growing need to develop cost-effective methods for estimating soil properties by combining multiple spectral libraries scanned using different spectrometers at the local, regional and global scales.”

Comment 2: Important methodological developments should certainly present in the manuscript, but some routine analysis methods or procedures needn’t show in “materials and methods” section, just marked with links/references or directly move to SI section.

Response: Thank you. We have revised the methodology section to remove the description of routine analysis methods or procedures and provided appropriate citations.

Comment 3: Although the manuscript title is short and simply, however it was difficult to understand clearly and directly. Suggest the title rewriting in order to reflect the text and meaning of the manuscript.

Response: Thanks for the reviewer suggestion. We have revised the tile as follows:

“Is standardization necessary for sharing of a large mid-infrared soil spectral library?”

Minor:

  1. In materials and methods section, all subtitles should be renumbered. For instance, Line 155 should be changed as “2.2 Spectral preprocessing and calibration transfer”, Line 182 “2.3 …”, …, and etc.

Response: Thank you. We have corrected the subtitle numbers.

  1. Important literatures are cited, some of them were too old in your 53 references.

Response: We have updated the old citations with the new ones. However, in some instances, we have to keep the old references because some of the calibration transfer techniques originate from these papers.

Reviewer 3 Report

Dear authors,

The manuscript tackles an important subject of soil spectral standardization and calibration transfer. Contribution to new knowledge by the manuscript has efficiently spelt out, the Materials and Methods section is well-written, the Results and Discussion section is informative enough and the results have been well-discussed. However, some minor improvments need to be done in Abstract and Introduction before the publication of the manuscript.

- The Abstract is not well informative and more details need to be provided. For instance, Please clarify what the European test set is (Ln. 20). Which "soil properties" (Ln. 25)? Which spectral preprocessing tecniques did you use?

- Introduction provides sufficient information with a suitable flow. However, gap of the knowledge is lacking and needs to be specified.

- Methodology has been adequately described with enough details.

- Results have been provided and interpreted very well with a strong discussion. However, I do not know why the Tables and Figures have not been included in the text, but provided as suplemantary materials. 

In general, I found this manuscript worthy of publication after an imperative and minor revision.

Author Response

The manuscript tackles an important subject of soil spectral standardization and calibration transfer. Contribution to new knowledge by the manuscript has efficiently spelt out, the Materials and Methods section is well-written, the Results and Discussion section is informative enough and the results have been well-discussed. However, some minor improvments need to be done in Abstract and Introduction before the publication of the manuscript.

Comment 1: The Abstract is not well informative and more details need to be provided. For instance, Please clarify what the European test set is (Ln. 20). Which "soil properties" (Ln. 25)? Which spectral preprocessing tecniques did you use?

Response: We thank the reviewer for the comments. We have revised the abstract to include the description on European test set and also provided the list of soil properties used for prediction with and without calibration transfer.

Comment 2: Introduction provides sufficient information with a suitable flow. However, gap of the knowledge is lacking and needs to be specified.

Response: In the introduction, we have clearly provided information on different calibration transfer techniques and their limitation. There has been mixed results on the performance of predictive models with and without calibration transfer. Therefore, we try to address this issue by using the PDS techniques across samples of different origin to examine whether calibration transfer is necessary. We have revised the final paragraph of the introduction section to include the knowledge gap as provided below:

“Although considerable effort has been placed on merging spectral libraries and developing robust predictive models using diffuse reflectance spectroscopy, it is still unclear whether predictive models built using soil samples scanned on a primary instrument can be directly applied to soil samples scanned on the secondary instrument.”

Comment 3: Methodology has been adequately described with enough details.

Response: Thank you

Comment 4: Results have been provided and interpreted very well with a strong discussion. However, I do not know why the Tables and Figures have not been included in the text, but provided as suplemantary materials. 

Response: We appreciate the reviewer comments. We have provided figures relevant to the main findings of the study (Fig 2, 3, 4) as the text. The tables and figures in the supplementary materials mostly support the findings related to figs 2, 3 and 4.

In general, I found this manuscript worthy of publication after an imperative and minor revision.

Reviewer 4 Report

The manuscript presents an extensive and well motivated study of benefits coming from calibration transfert by Piecewise Direct Standardization (PDS) in soil analysis by MIR spectroscopy.

The study compares predictions obtained by means of a secondary spectrometer, with and without PDS, with those obtained from primary spectrometer (i.e. that used in calibration). Two pre-treatment methods, three prediction algorithms and five target variables were considered. Two independent valiadation sets were used to test the results.

The study is, therefore,quite extensive and interesting for all reserchers working in spectroscopy field.

Presentation is generally clear, reference list and graphic material (included supplementary plots and tables) is adequate. I have detected only few unclear points requiring some fixing, which are listed below. Therefore, only a minor revision is required.

The reasoning at lines 46-48 is not very clear. I agree that impact of particle size increases at shorter wavelengths, but I do not understand why such issue should be more acute in the MIR (which has longer wavelengths) than in the NIR. The concept shoud be rephrased.

There is a likely typographic error at line 67. Maybe "from the secondary spectrometer", instead of "form", was intended.

At lines 87-88 derivative pre-processing is suggested reducing noise method. It looks a bit odd. It is true that derivative remove baseline shift, but, in my experience, derivative spectra tend to be more noisy than original ones, even using smoothing windows. Such point should be addressed more clearly.

Another unclear point is at lines 186-187 (outlier detection). What exactly mean "to pick up the maximum of 1% of poorly performing samples". That at most 1% of the whole dataset is removed? Or that only 1% of poorly performing samples is removed? In the latter case there should be two thresholds: one for defining poorly performing samples and one for defining the effectively removed samples. I have also read Ref. [15], but it is not more clear. I suggest to rephrase this concept more clearly.

Author Response

The manuscript presents an extensive and well motivated study of benefits coming from calibration transfert by Piecewise Direct Standardization (PDS) in soil analysis by MIR spectroscopy.

The study compares predictions obtained by means of a secondary spectrometer, with and without PDS, with those obtained from primary spectrometer (i.e. that used in calibration). Two pre-treatment methods, three prediction algorithms and five target variables were considered. Two independent valiadation sets were used to test the results.

The study is, therefore,quite extensive and interesting for all reserchers working in spectroscopy field.

Presentation is generally clear, reference list and graphic material (included supplementary plots and tables) is adequate. I have detected only few unclear points requiring some fixing, which are listed below. Therefore, only a minor revision is required.

Comment 1: The reasoning at lines 46-48 is not very clear. I agree that impact of particle size increases at shorter wavelengths, but I do not understand why such issue should be more acute in the MIR (which has longer wavelengths) than in the NIR. The concept shoud be rephrased.

Response: Thank you for pointing out this issue. We have removed lines 46-48 and focused the study on differences in spectral responses due to environmental conditions and instrumental differences.

Comment 2: There is a likely typographic error at line 67. Maybe "from the secondary spectrometer", instead of "form", was intended.

Response: Thank you. We have corrected the typographic error.

Comment 3: At lines 87-88 derivative pre-processing is suggested reducing noise method. It looks a bit odd. It is true that derivative remove baseline shift, but, in my experience, derivative spectra tend to be more noisy than original ones, even using smoothing windows. Such point should be addressed more clearly.

Response: We agree with the reviewer. Our point was that transformation can help to reduce noise coming from the instrument and help to develop robust calibration transfer matrix by removing baseline shifts. We have revised the text to clarify that transformation can help to develop better PDS models but not necessarily reduce noise.

Comment 4: Another unclear point is at lines 186-187 (outlier detection). What exactly mean "to pick up the maximum of 1% of poorly performing samples". That at most 1% of the whole dataset is removed? Or that only 1% of poorly performing samples is removed? In the latter case there should be two thresholds: one for defining poorly performing samples and one for defining the effectively removed samples. I have also read Ref. [15], but it is not more clear. I suggest to rephrase this concept more clearly.

Response: Thank you for the reviewer suggestion. We actually removed 1% of the samples from the whole datasets that are poorly performing. To accomplish this, we used a standard deviation threshold to pick the maximum of 1% of the poorly performing samples. The poorly performing samples are the samples that are farthest away from the 1-to-1 line. We have rephrase the text to clarify that we removed the maximum of 1% of the samples using the standard deviation threshold.

In the first step, we detected outliers from the calibration set scanned using KSSL spectrometer by building PLSR models and testing their performance against all samples in the spectral library. We optimized outlier detection using a standard deviation threshold (0.1 to 3 at increments of 0.02) from the 1-to-1 line to pick up the maximum of 1% of the samples as outliers [13]. To accomplish this, we picked samples that are farther away from the optimized standard deviation threshold.”